# Driver Activity Classification Using Generalizable Representations from Vision-Language Models

## Abstract

*Driver activity classification is crucial for ensuring road safety, with applications ranging from driver assistance systems to autonomous vehicle control transitions. In this paper, we present a novel approach leveraging generalizable representations from vision-language models for driver activity classification. Our method employs a Semantic Representation Late Fusion Neural Network (SRLF-Net) to process synchronized video frames from multiple perspectives. Each frame is encoded using a pretrained vision-language encoder, and the resulting embeddings are fused to generate class probability predictions. By leveraging contrastively-learned vision-language representations, our approach achieves robust performance across diverse driver activities. We evaluate our method on the Naturalistic Driving Action Recognition Dataset, demonstrating strong accuracy across many classes. Our results suggest that vision-language representations offer a promising avenue for driver monitoring systems, providing both accuracy and interpretability through natural language descriptors. We make our code available at [anonymized].*

## 1. Introduction

Distracted driving is a common factor in many vehicle accidents [1]. Systems which monitor the driver can offer advisories to the driver which encourage maintained focus on the road [2–7]. These advisories can be effective at reducing the occurrence or severity of related accidents [8].

Another solution to individual transportation lies in autonomous vehicles, with a distant goal that distracted driving is no longer a problem if the person in the driver's seat is not expected to be controlling the vehicle. However, current systems encounter failure cases and novel scenarios [9, 10]. Systems cannot safely transfer control without awareness of the driver, as the driver may be sleeping or pre-occupied with a distracting activity. For this reason, in-cabin driver monitoring and understanding of the driver state is critical for control transitions in autonomous systems too [11–13].

## 2. Related Research

Models which treat driver monitoring as a closed-set task [14, 15] have found success on benchmark datasets [16, 17].

However, the real environment is open-set [18, 19]. While our provided method is not open-set in its training data, by using a foundation model backbone, the encoding network has already learned representations of nearly any activity class. This makes the method highly adaptable to any visual activity class, though learning to classify those encoded representations may still require closed-set supervised learning (or, an unsupervised or active method to identify novel classes [20]) to provide desired predictions suitable to the open-set world.

Further, the real environment contains drivers which are out-of-distribution for a fixed set of training subjects. This is a problem when using data-driven methods which are tuned based on visual features. Some solutions lie in abstractions which remove the driver identity from the image [21, 22]. Related to this problem is the challenge of generalizing to drivers without training data; for most situations, it is infeasible that the vehicle monitoring system capture input of the driver, annotate this input, and use it to finetune a system. This motivates the need for zero-shot learning, where the system is expected to perform with zero prior training instances of the test subject [23, 24].

In this research, we introduce a method which represents the driver in a language-based visual descriptor. Though this representation utilizes image-based features, the features are learned in relationship to verbal descriptors, which pushes the representation from one based purely on pixel values to a representation which is based on the meaning of patterns found in those pixels, to the extent that they can be described by natural language.

# 3. Methodology

## 3.1. Algorithm

Our algorithm for driver activity classification is presented in Algorithm 1, including encoding, network approximation, and post-processing.

---

**Algorithm 1:** SRLF Activity Classification Algorithm

---

**Input:** Synchronized video frames
**Output:** Filtered probabilities per instance
**foreach** *triplet of frames* **do**
    **foreach** *frame* **do**
        Create an embedding for the image using the CLIP pretrained vision encoder;
    Pass the three embeddings as input to the SRLF neural network;
    Take argmax over output to receive single class probability per frame;
    Apply a mode filter with window size $w$ over the resulting probabilities;

---

We use $w = 141$ for our inference data sampled at 30 Hz, but this parameter should be tuned to match the typical duration of the driver activities, relative to the rate at which the network generates predictions or processes input.

## 3.2. Semantic Representation Late Fusion Neural Network

Our network, Semantic Representation Late Fusion Neural Network (SRLF-Net) is presented in Figure 1. The network consists of $N = 3$ parallel CLIP ViT image encoders, followed each by an FCN encoder, after which the outputs of the $N$ tracks are fused before entering a deep FCN network to generate class probability output.

## 3.3. Leveraging Generalizable Representations from Language-Vision Foundation Models

With this representation, rather than the descriptor of each driver being a specific array of pixels which may represent that driver's facial structure, hairstyle, skin color, size, and other non-relevant traits, the information bottleneck and pretraining mechanism instead reduce the amount of information and preserve (at least, to the ability of the optimizer) only features which are useful in organizing the images in a latent space that is separable by language. Of course, it is possible that with language we can describe concepts like facial structure, hairstyle, skin color, etc., but what is important is that the verbal description of these properties is a much lower amount of information than having the complete set of pixels which define that facial structure or hairstyle or skin color. With this representation, it is our hypothesis that the model becomes significantly more generalizable when trained, as it loses its ability to overfit to the very-individual properties of specific drivers.

Taken to an extreme, we can view the act of classifying an image as a reduction to the minimal number of bits to represent the information we care about from an image. With this in mind, we can view the image itself as the representation with the most information (which may be more than is required to solve the problem, containing both noise and irrelevant detail), and the class itself as the most compact. It is possible to use a large language model to directly output a prediction of a class, but this relies on the tuning of many components, in particular, the text encoder of the classification-request prompt and the associated prompt phrasing, and the ability of the model which learns to decode the image to a class according to this prompt. In our results, we show that current large language models struggle to learn this task satisfactorily. Because the image encoding representation is a less-reduced representation, we suggest that this can be used as an intermediate (not too large, not too small) representation of the relevant information, from which we can learn appropriate patterns without requiring the tuning of a text encoder or the finetuning of the model parameters which connect a prompt encoding and vision encoding, significantly reducing computational and data requirements while still maintaining the necessary level of information to solve the classification problem.

## 3.4. Separating Visual and Semantic Information using Order-based Augmentation

While we would ideally extract a semantic-level representation of the images and remove the ability to overfit to pixel configurations, the CLIP representation still carries some image features forward. However, we introduce a method which mitigates the overfitting possibility by a specialized data augmentation. If we treat the order in which the three views are passed to the network as random, we may be able to push the model to learn features within the 768-vector which represent semantic information as opposed to image-feature information, since the image-feature information would vary for each view while the semantic feature information should remain consistent. This method can also be extended to any number of views. While an overparameterized model may simply learn additional representations (for each permutation of image order), an appropriately-parameterized model may show better generalizability performance through this semantic-pass-filter information bottleneck.

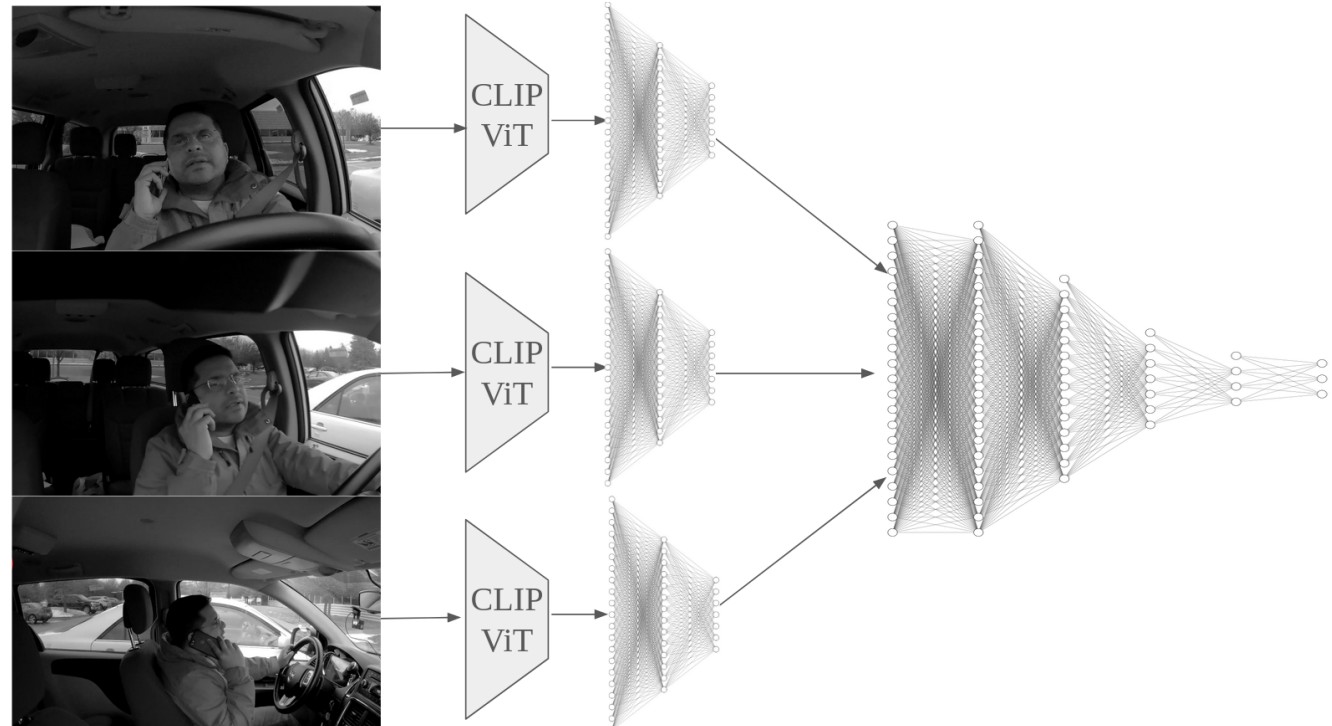

Figure 1. The Semantic Representation Late Fusion Network (SRLF-Net) takes images from multiple perspectives as input. Each image is sent to a CLIP encoder. Our experiments use the Vision Transformer backbone, base size, with size 32 patches. These representations are then further encoded using independent (non-shared-weight) fully-connected layers, each followed by batch normalization, ReLU activation, and dropout (rates 0.5 and 0.6 respectively). We use input size 768, and use two layers, compressing once to 512 and then to 256. These representations are then concatenated and used as input to another series of fully-connected layers (fusion step), again using batch normalization and ReLU activation between each. The size of these layers are 768, 768, 512, 256, 128, then $n$ (number of classes), which is 16 for our experiments.

## 4. Experimental Evaluation

### 4.1. Dataset

We utilize the Naturalistic Driving Action Recognition Dataset from the AI City Challenge [25], which consists of approximately 62 hours of footage, acquired from 69 participants. Each participant performed 16 different tasks, including but not limited to telephonic conversations, eating, and reaching backward, in a randomized order, as specified in Table 1.

The data includes three camera positions installed within a vehicle, as in Figure 2, positioned to capture from varied angles and synchronized to record simultaneously. The data collection was executed in two phases for each participant: the first without any visual obstructions and the second incorporating visual obstructions to appearance (e.g., sunglasses, hats). Thus, six videos were collected per participant—three from the non-obstructed phase and three from the obstructed phase.

| Class | Activity Label | Dist. % |
|---|---|---|
| 0 | Normal Forward Driving | 59.01 |
| 1 | Drinking | 1.49 |
| 2 | Phone Call(right) | 2.78 |
| 3 | Phone Call(left) | 2.97 |
| 4 | Eating | 3.29 |
| 5 | Text (Right) | 3.44 |
| 6 | Text (Left) | 3.56 |
| 7 | Reaching behind | 1.40 |
| 8 | Adjust control panel | 2.42 |
| 9 | Pick up from floor (Driver) | 1.31 |
| 10 | Pick up from floor (Passenger) | 2.15 |
| 11 | Talk to passenger at the right | 3.52 |
| 12 | Talk to passenger at backseat | 3.46 |
| 13 | Yawning | 1.87 |
| 14 | Hand on head | 3.45 |
| 15 | Singing or dancing with music | 3.85 |

Table 1. Table of Driver Activity Classifications.

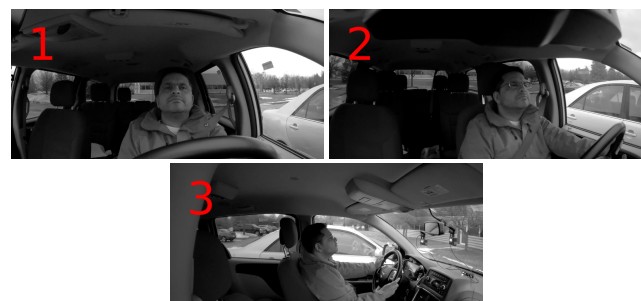

Figure 2. Illustration of multi-perspective in-cabin camera views for monitoring driver behavior under the class '0: Normal Forward Driving'. (1) Dashboard view. (2) Rear-view. (3) Side view.

## 4.2. Training Details

We detail our evaluation data splits in the following sections, with care to have images of individuals binned only to one set out of training and test. We divide our training set into two groups; 80% to train and 20% to validation, with possible overlap in individuals (though no same frames are shared). With our training set, we train SRLF-Net for up to 100 epochs, employing early stopping on a validation loss criteria. We use the adam optimizer (learning rate of 0.0001), 1cycle learning rate schedule policy [26], and cross-entropy loss.

For testing, we utilize the 7-fold data split provided in the dataset, dividing into 7 near-even groups of participants. This allows us to approximate generalizability with a 7-fold average.

## 4.3. Evaluation Over All Classes

The results for 7-fold test are seen in Table 2. We achieve an average accuracy of 71.64 %, showcasing the promising use of the method, notable in comparison to 6.25% expected accuracy of random selection for sixteen classes.

| K-fold | Accuracy |
|---|---|
| 1 | 68.09 % |
| 2 | 74.40 % |
| 3 | 73.60 % |
| 4 | 71.37 % |
| 5 | 70.15 % |
| 6 | 75.34 % |
| 7 | 68.53 % |
| **Average:** | 71.64 |
| **Standard Deviation:** | 2.88 |

Table 2. Table of k-fold cross-validation accuracies and average accuracy.

As illustrated in Figure 3, the model observes a large favorability for class 0 (Normal Forward Driving) likely due

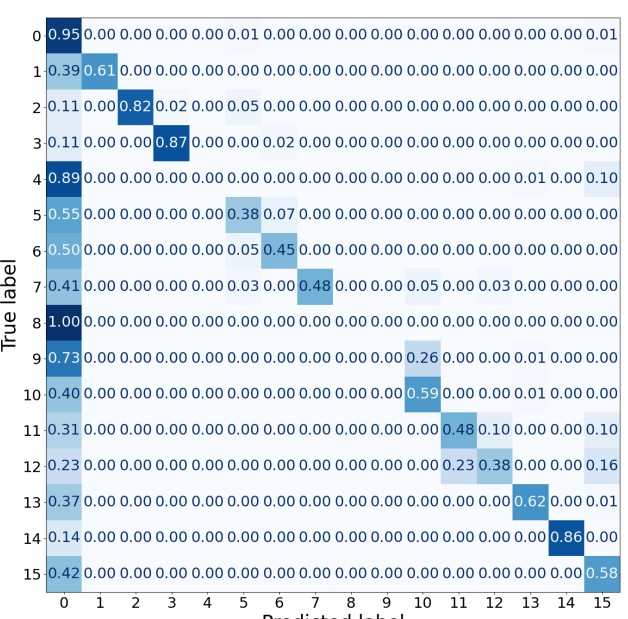

Figure 3. Confusion matrix for best performing k-fold 6 including a mode filter, resulting in a performance of 77.10 %.

to the skewed distributions of the data, as portrayed in Table 1, with phone call and hand-on-head the next most-correctly-classified classes. Adjusting the control panel shows the most confusion with the default driving class. Straight forward driving accounts for 59.01 % of the data, resonating binary test to differentiate between straight forward driving and all other classes in Figure 4. For more accurate classification, it would be beneficial to mitigate the effects of the confounding majority class ("normal driving"); we explore experiments in class-weighting, but find these effects to not be strong enough to counter the adverse learning effect. As another solution, we consider the use of an early-stage binary classifier to separate normal driving from distracted driving. The binary classifier is imperfect (as shown in Figure 4, and in the next section, we carry out an additional distraction-classification experiment excluding the "normal driving" class, on the assumption that some strong binary classifier may be achieved with further architectural exploration.

## 4.4. Distracting Activities Only: Evaluating Without Normal Driving Class

Our architecture, in combination with a dataset heavily skewed towards normal driving, tends to overpredict the normal driving class. To understand how well the model separates between the distracting activity classes, we run an experiment by which we assume there is some "perfect" binary classifier which can distinguish between normal driving and distracted driving, and then use our model to classify only between these distraction classes. The results of

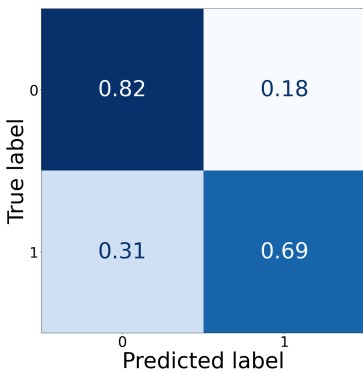

Figure 4. Binary Confusion matrix for best performing k-fold 6 only including class 0 for straight forward driving and a combination of all other activity classes, performing 77.22 % accuracy.

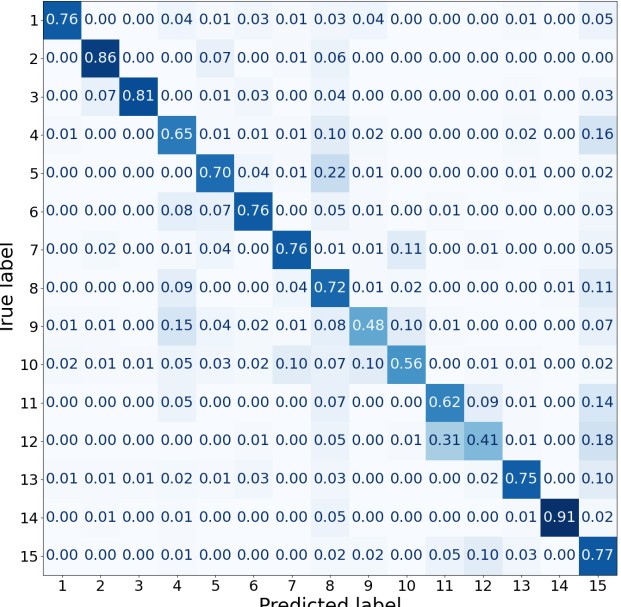

Figure 5. Confusion matrix for best performing k-fold 6 without class 0 for straight forward driving and including a mode filter, performing 70.06% accuracy. By removing the forward driving class, the accuracy metric decreases slightly (simply because the over-predicted forward driving class accounted for a majority of the dataset), but the average performance over classes actually increases from 50.44% to 70.13%. The alignment of average per-class accuracy and overall accuracy is a strong indicator of the model's effective learning.

this experiment are illustrated in Figure 5. The model, in general, predicts the correct class with the greatest likelihood for any given activity class, though for some individual classes, this likelihood may be less than >50%. Phone call and hand-on-head again show the best performance.

We also highlight the importance of the mode-filter post-processing step; without the mode filter, the accuracy is 63.66%, and with the mode filter, this accuracy rises to 70.06%. This filter leverages the knowledge that there is a certain rate at which a driver can reasonably change between tasks (i.e. it would be unexpected for a driver to oscillate between different distracting activities at 30 Hz, even if the camera captures and model infers at that rate).

# 5. Concluding Remarks and Future Research

To begin, we highlight some recommended opportunities for future research:

1. Comparison to text-encoding methods, such as vector products between text and image encodings, or even the evaluation of prompted vision-language systems to determine classes of images. We note that we have began a series of experiments using LLaVA, but the computation time on such methods *significantly* exceeds the method shown in this paper, without offering stronger preliminary results. In relation to these methods, our presented algorithm does carry the benefit of immediate applicability to multiple simultaneous views.

2. The integration of temporal information (either as post-processing, or addition of LSTM or Transformer models early in the architecture) may be very useful, since driver activities occur over time, with valuable information in these action dynamics.

3. Evaluation on combinations of non-consistent views. It would be interesting to merge multiple datasets which share some classes in common, so that we can evaluate generalizability to further views and subjects.

4. Integration into open-set novelty detection methods, such that the system can expand its number of classes, retraining if necessary, when new activities are introduced.

In this research, we present a new perspective of the vision-language contrastively-learned encoding as a fundamental new representation of an image, which contains both visual information as well as semantic information. We show that from this information, it is possible to classify driver activity into a variety of distraction classes with fairly strong accuracy, and further, that our algorithm can adapt to any number of simultaneous views. Vision-language models may lead to driver monitoring systems which are more accurate, robust, and generalizable; suitable for an open-set of possible distractions; and directly explainable [27] via language.

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
