# OpenReview forum: "Driver Activity Classification Using Generalizable Representations from Vision-Language Models"
_thecvf.com/CVPR/2024/Workshop/VLADR — VLADR 2024 Poster_

### Official Review · Reviewer_QukY · 2024-04-20
**Application of VLM on DMS**

**Rating:** 6
**Confidence:** 3

**Review:**

The paper presents a driver activity classification for road safety and autonomous vehicle development. The authors propose a Semantic Representation Late Fusion Neural Network (SRLFNet) that leverages generalizable representations from vision-language models. The method processes synchronized video frames from multiple perspectives, encoding each frame with a pretrained vision-language encoder and fusing the embeddings to predict class probabilities. The novelty of model architecture is limited, and I suggest further exploration.

---

### Decision · Program_Chairs · 2024-04-22

Accept (Poster)